# Distinct transcriptional responses elicited by unfolded nuclear or cytoplasmic protein in mammalian cells

Yusuke Miyazaki, Ling-chun Chen, Bernard W Chu, Tomek Swigut, Thomas J Wandless*

Department of Chemical and Systems Biology, Stanford University, Stanford, United States

**Abstract** Eukaryotic cells possess a variety of signaling pathways that prevent accumulation of unfolded and misfolded proteins. Chief among these is the heat shock response (HSR), which is assumed to respond to unfolded proteins in the cytosol and nucleus alike. In this study, we probe this axiom further using engineered proteins called 'destabilizing domains', whose folding state we control with a small molecule. The sudden appearance of unfolded protein in mammalian cells elicits a robust transcriptional response, which is distinct from the HSR and other known pathways that respond to unfolded proteins. The cellular response to unfolded protein is strikingly different in the nucleus and the cytosol, although unfolded protein in either compartment engages the p53 network. This response provides cross-protection during subsequent proteotoxic stress, suggesting that it is a central component of protein quality control networks, and like the HSR, is likely to influence the initiation and progression of human pathologies.

## Introduction

The life of a cell is characterized by homeostasis, where a constant input of energy is required to balance the synthesis and degradation of the molecules that are essential for viability. This is a difficult balancing act under ideal conditions, but cells are often not so lucky. Rather, sources of stress ranging from radiation to exogenous chemical and biological agents and even chemical byproducts of primary metabolism can cause damage to critical molecules (*Balaban et al., 2005*). To counteract these insults, cells have evolved surveillance mechanisms to detect damage, allowing them time to repair the damage or to otherwise adapt to stress (*Stoecklin and Bukau, 2013*). For example, the integrity of the genome is important for cellular viability, and it is closely monitored (*Ciccia and Elledge, 2010*). When DNA damage is detected, one or more cellular checkpoints are activated to allow the cell time to repair the damage, or in the case of multicellular organisms, initiate apoptosis if the damage is too severe to repair (*Jackson and Bartek, 2009*). Similarly, the spindle assembly checkpoint ensures that mitosis does not proceed until each sister chromatid is productively engaged by microtubules at its kinetochore (*Foley and Kapoor, 2013*). Defects in either of these quality control (QC) surveillance mechanisms lead to genomic instability and frequently drive human pathologies such as cancer (*Sperka et al., 2012*).

Proteins are equally important to cellular viability. In the past several decades, studies in organisms from bacteria to humans have revealed that cells broadly monitor protein QC during translation as well as after mature proteins have entered the proteome (*Chen et al., 2011*; *Wolff et al., 2014*). Damaged proteins are typically degraded by either the ubiquitin-proteasome system (UPS) or through autophagy, although we still lack a clear understanding of how cells detect damaged proteins and how they decide to either refold or degrade these substrates if refolding is not productive

*For correspondence:
wandless@stanford.edu

**Competing interests:** The authors declare that no competing interests exist.

**eLife digest** The majority of cellular proteins must be folded into a precise shape to work correctly. But when cells experience stressful conditions such as high temperatures and harmful chemicals, proteins start misfolding or even completely unfold. Misfolded proteins can be highly toxic. As a result, cells use several different groups of molecules that work together in pathways to both detect and then refold incorrectly folded proteins.

Protein misfolding has mainly been studied under extreme experimental conditions that cause hundreds of different proteins to misfold at the same time. In 2006, researchers developed a technique that can be used to switch a protein between a folded and an unfolded state. Now, Miyazaki et al.—including several of the researchers involved in the 2006 work—have used this technique to investigate how a cell responds when a single protein unfolds. The sudden appearance of the unfolded protein increased the expression of several genes. Some of these genes are part of known pathways, but the majority were not known to respond to protein-folding stress. The cell also reacts differently depending on whether the unfolded protein is made in the cell nucleus, or the cytosol (the fluid that surrounds the nucleus and other cellular components).

Miyazaki et al. discovered that a network of molecules controlled, in part, by a protein called p53 activates the response to unfolded proteins in either the nucleus or the cytosol. Cells that activated this network were also better able to protect themselves from further stress. This newly discovered pathway is likely to play an important role in monitoring the quality of the proteins made in the cell. The next step involves identifying the cellular sensor(s) that recognize the unfolded protein as well as the transcription factors that are responsible for increasing gene expression. With a more complete picture of this new response, it will be possible to knock-out specific pathways to determine their roles.

(*Wong and Cuervo, 2010*). However, it is clear that disruptions to the protein-folding environment trigger cellular stress responses that mitigate the effects of the misfolded or unfolded proteins (*Vabulas et al., 2010*; *Walter and Ron, 2011*; *Nargund et al., 2012*). Not surprisingly, the inability to mount these stress responses is correlated with decreased cellular fitness, especially under conditions of protein-folding stress (*Morimoto, 2011*). Failure to refold or degrade unfolded or misfolded proteins is detrimental to the health of cells and organisms and is linked to a multitude of human diseases (*Morimoto, 2008*; *Knowles et al., 2014*).

The first of the protein stress responses to be described was the heat shock response (HSR) in 1962, although it was another 20 years before the sensor, the transcription factor HSF1, was identified (*Ritossa, 1962*; *Parker and Topol, 1984*; *Wu, 1984*). The HSR provided the first molecular mechanism for a cellular response to proteotoxic stress, in this instance presumably triggered by thermally unfolded proteins. A decade passed before Walter and colleagues discovered a second unfolded protein response activated by misfolded proteins in the secretory pathway (*Cox et al., 1993*). This unfolded protein response, called the erUPR (Unfolded Protein Response), induces the transcription of genes involved in helping proteins fold or otherwise restoring the secretory pathway to a state that is conducive to protein homeostasis (*Walter and Ron, 2011*). The erUPR also resets protein translation, reducing the synthesis of additional proteins when the environment is not conducive to proper folding (*Harding et al., 2000*). More recently, Ron and Haynes have characterized a conceptually similar unfolded protein response in mitochondria (mitoUPR) (*Haynes et al., 2007*).

One common feature of these three protein-folding stress responses is that they induce the transcription of genes involved in mitigating the effects of stress. The HSR induces protein chaperones to assist in refolding (*Vabulas et al., 2010*). The erUPR induces the production of ER-resident chaperones as well as biosynthetic genes that lead to the expansion of the ER, and the mitoUPR induces the transcription of Hsp60 and other nucleus-encoded mitochondrial genes to maintain homeostasis (*Walter and Ron, 2011*; *Pellegrino et al., 2014*).

A second common feature of these stress response pathways is that they are typically studied using extreme, often non-physiological perturbations such as transferring a plate of cultured cells to a higher temperature or treating cells with high concentrations of Dithiothreitol (DTT) or other drugs

that dramatically disrupt the folding environment in the secretory pathway (e.g., tunicamycin or thapsigargin) or in the mitochondria (e.g., ethidium bromide) (*Oslowski and Urano, 2011*; *Nargund et al., 2012*). All of these perturbations result in the misfolding of hundreds or thousands of different proteins and likely other biomolecules as well. Nevertheless, despite the indiscriminate nature of these perturbants, they have been extremely useful in allowing us to discover and study these fundamental stress response pathways.

Our interest in this area was sparked by a simple question: are mammalian cells sensitive to the appearance of one, specific unfolded protein? And if so, does the materialization of this unfolded protein elicit a response that helps cells eliminate or adapt to this stressor? We used engineered proteins whose folding state can be controlled using a cell-permeable small molecule to rapidly create a single unfolded protein in mammalian cells, and herein, we report that the appearance of unfolded protein elicits a co-ordinated transcriptional response that is distinct from the HSR and the other UPRs. Cells that trigger this response are more resistant than unstressed cells when challenged with other protein-folding stressors. Finally, the creation of unfolded protein in the nucleus elicits a response that is distinct from the response to unfolded cytosolic protein.

## Results

### Creating a single unfolded protein

We previously developed a technique to regulate the expression level of any protein-of-interest in living cells using a cell-permeable small molecule (*Banaszynski et al., 2006*). We started by screening a library of genes encoding mutants of the human FKBP12 protein to identify mutants whose metabolic stability could be regulated by a high-affinity ligand called Shield-1 (S1). These mutants are called destabilizing domains (DDs). In the absence of S1, the DDs are ubiquitylated and degraded by the proteasome, whose processivity ensures the degradation of any fused partner proteins (*Egeler et al., 2011*). We have similarly engineered *Escherichia coli* dihydrofolate reductase (ecDHFR) and the ligand-binding domain of the human estrogen receptor to behave as DDs using ligands such as trimethoprim and tamoxifen to regulate protein stability (*Iwamoto et al., 2010*; *Miyazaki et al., 2012*). Others have begun to apply these tools to study a wide range of biological processes (*Raj et al., 2013, 2014*; *Beck et al., 2014*; *Razooky et al., 2015*).

The DDs are not well folded in the absence of their stabilizing ligands (*Egeler et al., 2011*). NMR spectroscopy of several FKBP-derived DDs revealed that their capacity to induce degradation in cells correlates with their degree of unfolding in vitro. Complementary urea denaturation studies revealed that DDs sample unfolded conformations to different extents, but all of the DDs are strongly stabilized by the addition of S1. These mechanistic studies suggest that DDs extensively sample an unfolded conformational state when expressed in cells. Importantly, this unfolded conformation does not irreversibly aggregate, but rather equilibrates between unfolded and folded states, permitting S1 to stabilize the folded conformation. Unfolded DDs are recognized by the cellular QC machinery and targeted for degradation through ubiquitylation (*Egeler et al., 2011*; *Chu et al., 2013*). However, S1 binding prevents the DDs from sampling the unfolded state, thus, rescuing DDs from degradation. Therefore, we use S1 as a small molecule switch to toggle DDs expressed in cells between folded and unfolded states with a high degree of temporal control.

We took advantage of this unique conditional behavior to characterize the response mounted by mammalian cells to a single unfolded protein species. We stably introduced cDNA encoding the FKBP-derived L106P DD fused to superfolder Green Fluorescent Protein (GFP) (DD-GFP) fusion protein into NIH3T3 fibroblasts (*Pédelacq et al., 2006*). Fluorescence-activated cell sorting (FACS) was used to select those cells expressing high amounts of DD-GFP fusion protein. We maintained the stabilizing S1 ligand in culture media at all times to avoid either stressing the cells with unfolded protein or forcing the cells to adapt to the presence of unfolded protein. To initiate the stress, S1 was withdrawn to create the unfolded DD (*Figure 1A*). We harvested cells 45, 135, and 405 min following S1 withdrawal, isolated mRNA, and analyzed changes in the transcriptome using mRNA-seq. Changes in transcription are a hallmark of the known protein homeostatic stress responses (*de Nadal et al., 2011*).

As an experimental control, samples of the DD-GFP-expressing cells were treated with tunicamycin to initiate the UPR in the secretory pathway (erUPR), and cells were harvested at the same timepoints for mRNA-seq analysis. As an additional control for the HSR, DD-GFP-expressing cells were shifted

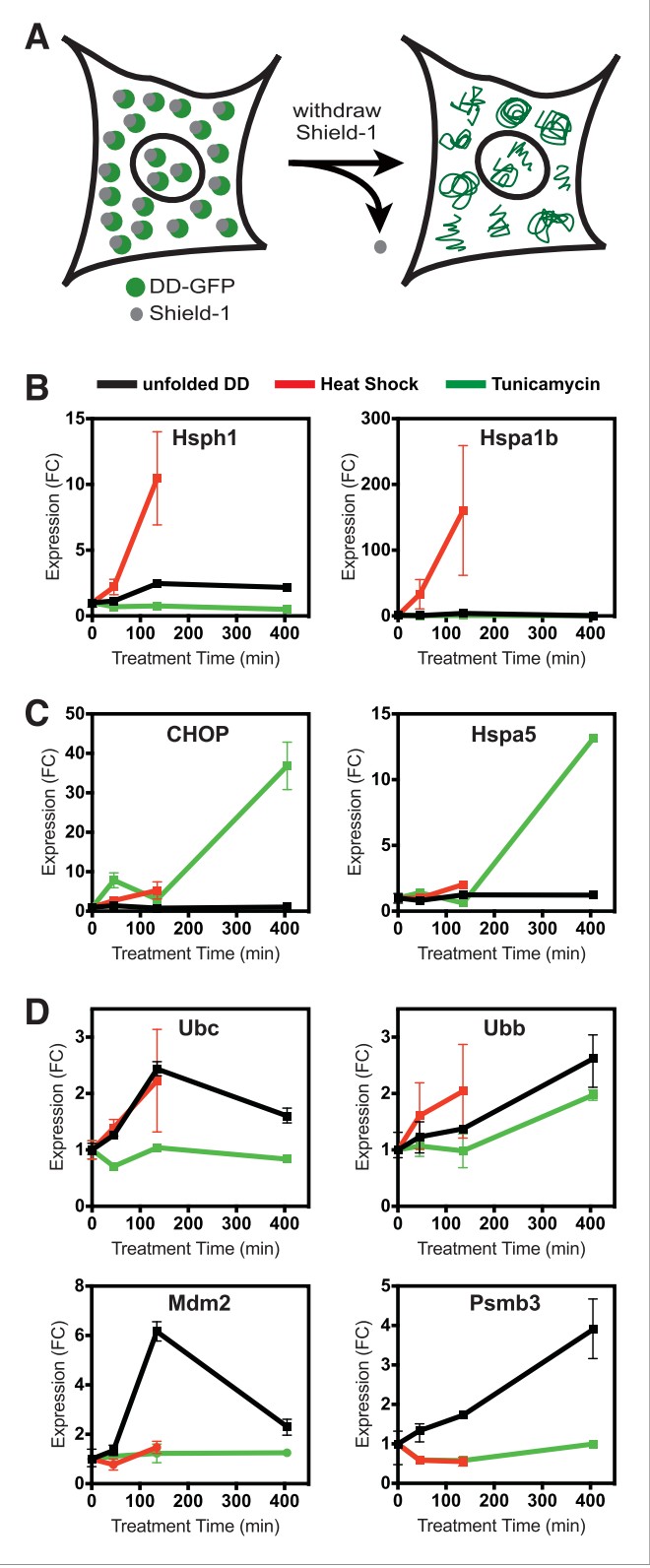

**Figure 1**. Unfolded DD induces transcriptional response. (**A**) Schematic representation of the strategy used to create a unfolded protein in cells using the destabilizing domain (DD). (**B–D**) Changes in transcript levels measured by mRNA-seq are shown for genes that respond strongly to heat shock (HS) (panel **B**), tunicamycin (panel **C**), and the appearance of the unfolded DD (panel **D**). Transcript levels are shown relative to unperturbed cells. FC = fold-change.

*Figure 1. continued on next page*

*Figure 1. Continued*

The following figure supplement is available for figure 1:

**Figure supplement 1**. Unfolded DD responsive genes from mRNA-seq.

from 37°C to 42°C and analyzed by mRNA-seq at 45 and 135 min. Heat-shocked cells were not analyzed at 405 min due to significant lethality. For both erUPR- and HSR-control samples, S1 was maintained throughout the experiment. The DD-GFP-expressing cells were exposed to the three conditions (unfolded DD, heat shock (HS), and tunicamycin), and mRNA-seq was used to quantify changes in transcript levels relative to the reference sample of DD-GFP-expressing cells in which the S1 ligand was not withdrawn. To identify any spurious effects that might arise from treating cells with S1, NIH3T3 cells were transduced with cDNA encoding superfolder GFP alone (*Maynard-Smith et al., 2007*). These cells were treated with S1 in the same manner as the DD-GFP-expressing cells, and mRNA-seq was performed at the same timepoints.

Cells incubated at 42°C showed strong induction of canonical heat-shock genes such as Hsph1 and Hspa1b (*Figure 1B*), confirming that this exposure causes them to mount an HSR (*Murray et al., 2004*). Similarly, treatment with tunicamycin induced expression of known erUPR genes such as *Hspa5/Bip* and *Ddit3/Chop* (*Figure 1C*) (*Murray et al., 2004*). Because DD degradation depends on ubiquitin, we examined changes in the mRNA levels of the four mammalian ubiquitin genes. Unfolded DD caused increased transcript levels of two ubiquitin genes, Ubc and Ubb, whereas expression of the other two ubiquitin-encoding genes was unchanged (*Figure 1D* and *Figure 1—figure supplement 1*). HS also led to increased levels of the *Ubc* mRNA, and all three perturbations caused increases in *Ubb* transcript levels. The mRNA-seq analysis also revealed genes whose mRNA levels were induced exclusively by the unfolded DD and not by the other two perturbations, including *Mdm2* and *Psmb3* (*Figure 1D*). A complete list of differentially expressed genes is provided in *Supplementary file 1* and *Supplementary file 2*. Quantitative real-time PCR (RT-qPCR) was used to validate the results from the mRNA-seq analysis as shown in *Figure 2* and *Figure 2—figure supplement 1A*.

To determine how different doses of unfolded protein impacted the induction of this cellular response, we created a second NIH3T3 cell line expressing the FKBP-derived DD fused to superfolder GFP, but with protein levels approximately threefold lower than the original level (*Figure 2—figure supplement 1B*). We also created a control cell line expressing unfused GFP in which GFP levels were similar to the DD-GFP cell line when stabilized by ligand. We cultured both the lower-expressing DD-GFP and the GFP-only cell lines in the presence of S1 and analyzed transcript levels using RT-qPCR following withdrawal of the stabilizing ligand. The control cell line expressing only GFP showed no significant changes in transcript levels. In contrast, cells expressing lower levels of DD-GFP displayed changes in mRNA levels of the same genes that were affected by higher levels of unfolded DD. Moreover, the degree of mRNA induction correlated with the level of protein expression (*Figure 2*, *Figure 2—figure supplement 1A*), establishing that lower levels of unfolded DD induced less pronounced changes in mRNA levels.

In order to test whether this transcriptional response is unique to the unfolded FKBP protein, we created a cell line expressing a DD derived from the *E. coli* DHFR protein fused to superfolder GFP (*Iwamoto et al., 2010*). Upon withdrawal of the stabilizing ligand, trimethoprim for this DD, we characterized the cellular response to the unfolded ecDHFR-derived DD using RT-qPCR (*Figure 2—figure supplement 1C*). The unfolded ecDHFR-derived DD elicited changes in mRNA levels similar to the changes induced by unfolded FKBP, although the extent of the changes was less pronounced.

## Global comparison of proteotoxic stresses

We used hierarchical clustering to compare the cellular responses to all three perturbations (*Figure 3A*), revealing one cluster of genes whose transcript levels are induced by the unfolded DD but not by the HSR or the erUPR (asterisk in *Figure 3A*). Analysis of this group revealed enrichment in genes involved in 'intracellular protein traffic' (p-value = 3.7e-6) and 'cell cycle' (p-value = 8.4e-6) when analyzed for biological process GO terms (*Huang et al., 2008*, *2009*). Pathway analysis of the same

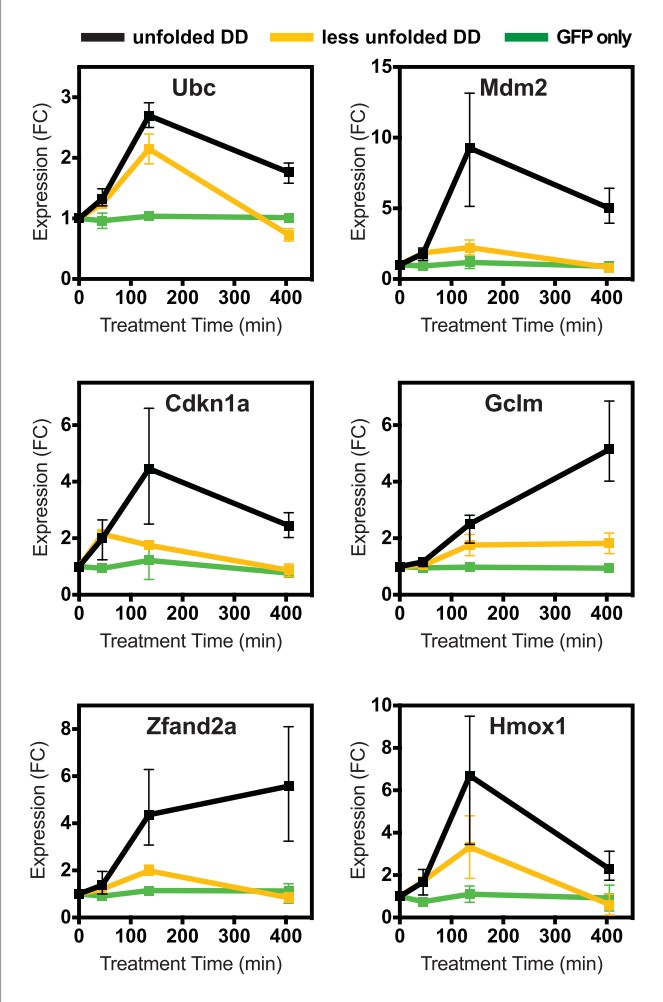

**Figure 2**. Cellular transcriptional response is dose-dependent. Unfolded DD was created in two cell lines, which express DD-GFP levels that differ by a factor of three (*Figure 2B*). Cells were harvested at the indicated times following withdrawal of Shield-1 (S1), and transcript levels were quantified using real-time PCR (RT-qPCR). Transcript levels were normalized to GAPDH and expressed relative to unperturbed samples.

The following figure supplement is available for figure 2:

**Figure supplement 1**. Lower levels of unfolded DD do not induce the stress response, but a different unfolded protein does.

gene cohort identified the 'ubiquitin-proteasome pathway' (p-value = 0.025) and the 'p53 pathway' (p-value = 0.041) as being enriched (*Supplementary file 3*).

In response to the unfolded DD, NIH3T3 cells up-regulated the expression of 0, 63, and 759 genes 45, 135, and 405 min, respectively, following withdrawal of S1 (false discovery rate [FDR] <0.05). The mRNA levels of 0, 69, and 909 genes were significantly reduced following S1 withdrawal (*Supplementary file 1* and *Supplementary file 2*). To put the magnitude of this response in perspective, NIH3T3 cells responded to HS by up-regulating the mRNA levels of 186 and 791 genes and down-regulating 26 and 579 genes at 45 and 135 min, respectively, following application of stress (*Supplementary file 1* and *Supplementary file 2*). Using tunicamycin to induce the erUPR, we observed increases in 586, 211, and 2403 and decreases in 349, 184, and 2659 transcript levels 45, 135, and 405 min following treatment (*Supplementary file 1* and *Supplementary file 2*). Given the potential for temporal differences in the cellular response to the different sources of proteotoxic stress, genes from all timepoints for each stress were grouped as either up-regulated

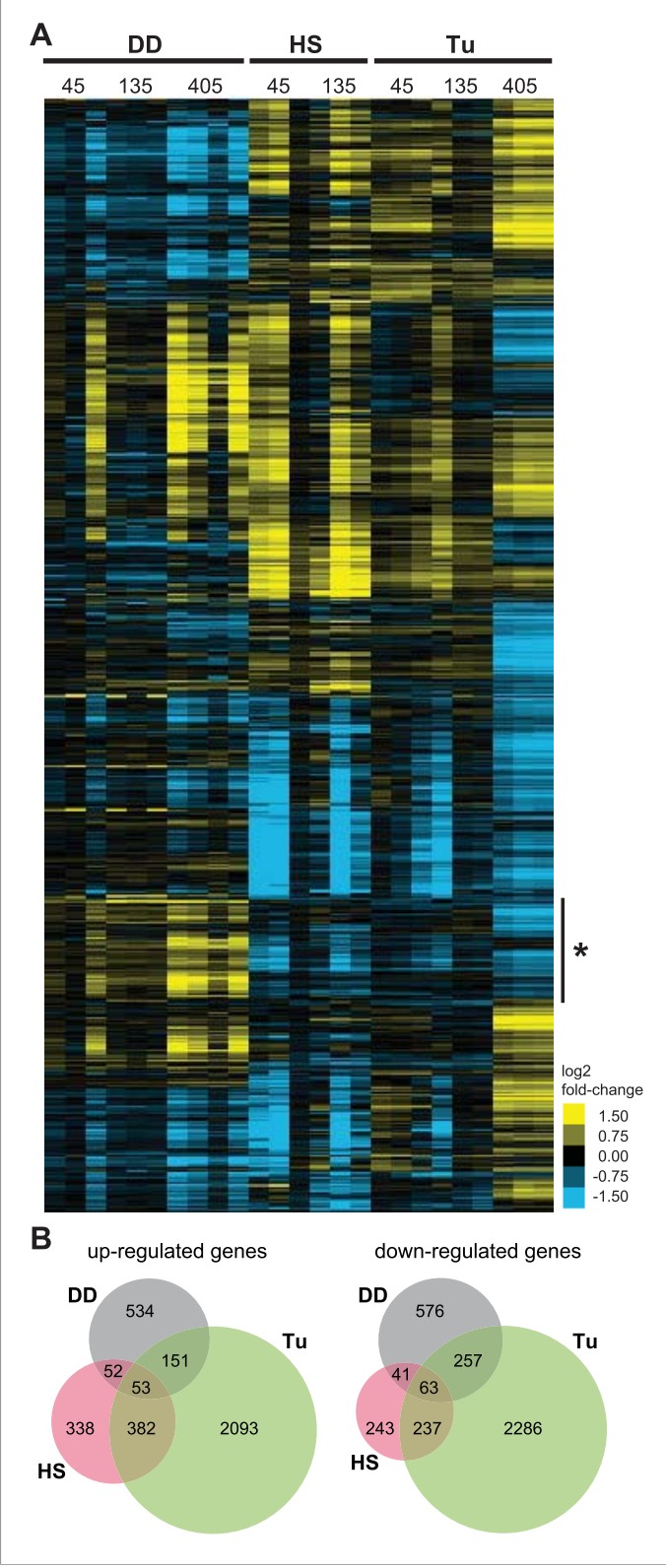

Figure 3. Transcriptome profiles of unfolded DD compared to HS and ER stress. (A) NIH3T3 cells expressing DD-GFP in the presence of S1 were perturbed by HS (42°C), treatment with 5 µM tunicamycin, or by withdrawal of S1 to create unfolded DD. Cells were harvested 45, 135, and 405 min following these perturbations, and transcript levels were quantified using mRNA-seq. Genes responding to any of the three perturbations (false discovery rate [FDR] <0.05) were hierarchically clustered to examine the relationships between the expression patterns. Each row
*Figure 3. continued on next page*

*Figure 3. Continued*

represents a gene and each column is an experimental replicate. Transcript changes are represented as log2 fold-changes relative to unperturbed samples. (**B**) Venn diagram depicting significantly induced and repressed genes (FDR <0.05) from each of the three treatments: unfolded DD, HS, and tunicamycin treatment (Tu).

or down-regulated, and the relationships among the three data sets are shown in Venn diagram format in *Figure 3B*.

Focusing on the 534 genes that are up-regulated in response to the unfolded DD but not the other two perturbations, gene enrichment analysis revealed that 'proteolysis' (p-value = 2.1e-4) and 'protein metabolism/modification' (p-value = 1.4e-3) are over-represented in the category of biological process. Pathway analysis revealed that the 'ubiquitin-proteasome pathway' (p-value = 9.3e-6) and the 'p53 pathway' (p-value = 0.021) are over-represented (*Supplementary file 3*).

Looking more closely at the responses of individual genes to the perturbations, we noticed that some mRNA levels rise rapidly whereas others are induced later. We then performed gene enrichment analysis for the unfolded DD samples using the list of earlier (mRNA levels rise significantly by 135 min) and later (405 min) responders. Analysis of the 63 early-response genes highlighted 'protein folding' (p-value = 7.5e-4) and 'stress response' (p-value = 2.1e-3) as the biological processes most significantly over-represented (*Supplementary file 4*). The same analysis for the late-response genes revealed 'protein folding', 'stress response', 'protein metabolism and modification', and 'proteolysis' as over-represented biological processes (p-values = 5.7e-6, 8.0e-6, 9.4e-6, and 4.9e-4, respectively, *Supplementary file 4*). When we analyzed these two gene groups for biological pathways, the early-response group was over-represented for the 'p53 pathway' and 'apoptosis-signaling pathway' (p-values = 1.2e-6 and 1.1e-3, respectively, *Supplementary file 4*). The late-response genes were enriched in 'ubiquitin-proteasome pathway' (p-value = 1.2e-4, *Supplementary file 4*). When analyzed for annotations related to molecular function, both groups revealed enrichment of 'Hsp70 family chaperone', 'chaperone', and 'microtubule family cytoskeletal protein' (p-values = 5.8e-4, 7.8e-4, and 3.3e-5, respectively, *Supplementary file 4*).

## New response is distinct from erUPR and HSR

We next compared the list of all genes induced by the unfolded DD against the list of all genes induced by either HS or by tunicamycin (*Figure 4A*). Only 105 genes overlapped between the 787 induced by unfolded DD and the 826 induced by HS (p-value = 3.00e-7 by hypergeometric comparison), suggesting that this degree of overlap is unlikely to have arisen by chance. Conversely, a similar comparison of the genes induced by unfolded DD with the erUPR genes induced by tunicamycin suggests that there is not significant overlap between the two cellular responses (*Figure 4A*).

The appearance of the 'p53 pathway' in our gene expression analysis led us to perform a comparison between the list of 790 genes induced by the unfolded DD and the 508 genes significantly up-regulated by p53 in response to DNA damage (*Figure 4A*) (*Kenzelmann Broz et al., 2013*). This analysis revealed 90 genes in common (p-value = 1.07e-13). Considered together, these results suggest that the acute appearance of unfolded DD elicits a co-ordinated transcriptional response in NIH3T3 cells. Gene expression analysis highlights biological processes and biological pathways associated with protein homeostasis and stress responses as over-represented in the lists of responsive genes. Furthermore, the cellular response to unfolded DD is distinct from both the known HSR and the erUPR in the secretory pathway.

Similarly, since the 'ubiquitin-proteasome pathway' appears in our gene expression analysis, we also treated cells expressing DD-GFP with the proteasome inhibitor, bortezomib, followed by mRNA-seq analysis at the same times as the other perturbations. Treatment with bortezomib induced the expression of 583 genes, with 65 in common with the unfolded DD perturbation. The overlap between these two sets of induced genes is not statistically significant (p-value = 2.42e-1), suggesting that the unfolded DD creates a cellular stress that is distinct from simple proteasome inhibition (*Figure 4—figure supplement 1*, *Supplementary file 5*).

We were concerned that the inclusion of statistically less significant genes induced by both perturbations made the cellular response to unfolded DD and the HSR appear to be dissimilar.

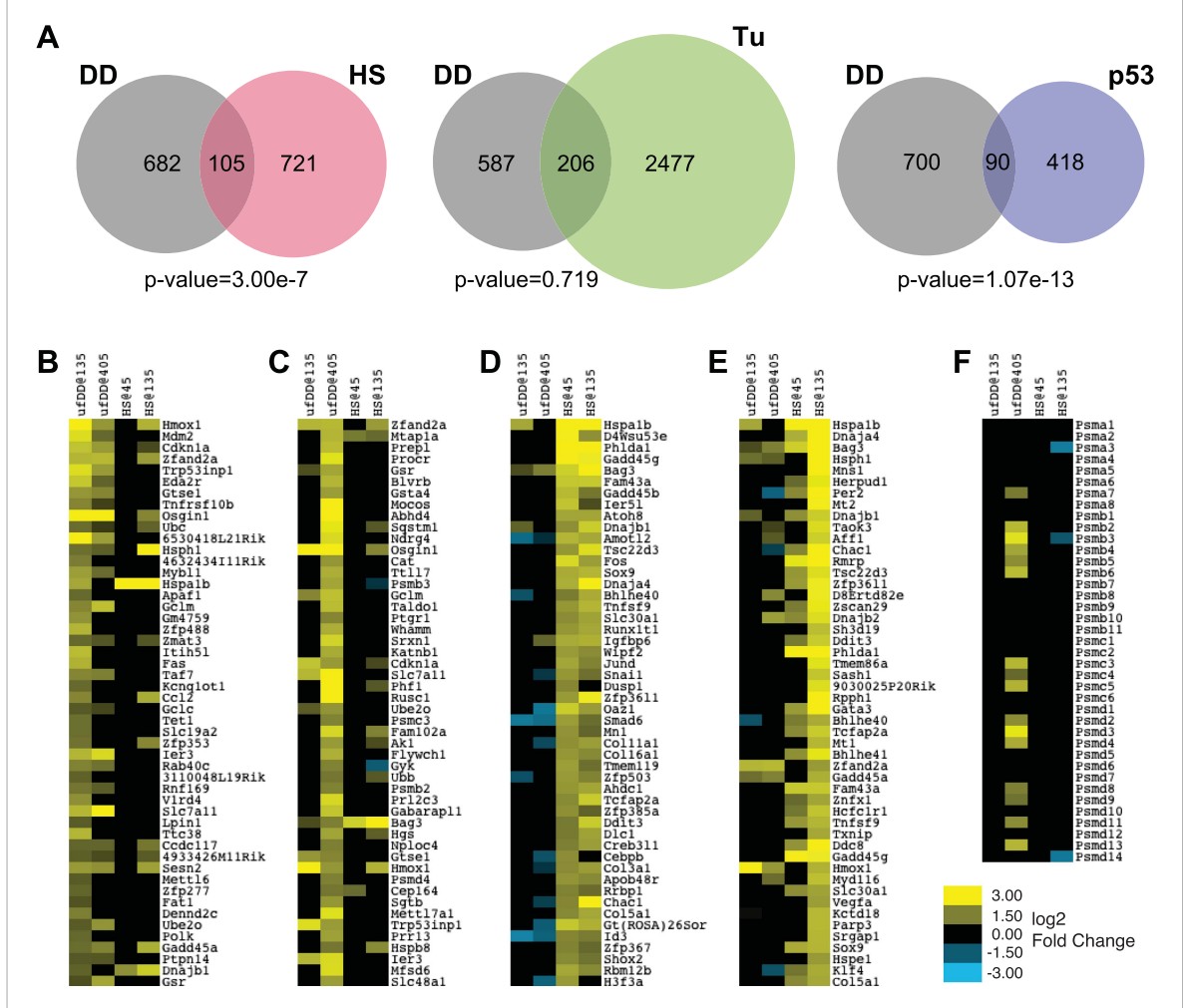

**Figure 4.** Comparison of unfolded DD stress with known stress responses. (**A**) Venn diagram showing pairwise comparisons of the genes induced by unfolded DD (gray) with the genes induced by the HS response (HSR) (red), the erUPR (green), and the p53-mediated response to DNA damage (blue). The list of p53-induced genes is from *Kenzelmann Broz et al. (2013)*. The p-values are calculated by hypergeometric test. (**B–E**) Heatmap representations of genes responding to unfolded DD or HS. The 50 most significantly induced genes are shown for each condition to facilitate comparison with the other perturbations and timepoints. The genes are sorted by statistical significance for the unfolded DD at 135 min (panel **B**) and 405 min (panel **C**) and for HS at 45 min (panel **D**) and 135 min (panel **E**). (**F**) Heatmap showing how 39 genes encoding proteasome subunits respond to either unfolded DD or HS. For panels **B–F**, black indicates no significant changes in transcript levels.

The following figure supplement is available for figure 4:

**Figure supplement 1.** Comparison of the unfolded DD stress with proteasome inhibition.

To explore this possibility, we plotted the 50 genes that were most significantly induced by either the unfolded DD or HS at both timepoints (*Figure 4B–E*). Looking at the 50 most significant genes induced by unfolded DD at the early timepoint, we find 28 of these genes are also significantly changed at the later timepoint, and all of these are induced (*Figure 4B*). However, comparison of the top 50 early genes induced by the unfolded DD with the HSR reveals only two significantly induced genes in common at the early HSR timepoint and 15 genes at the late HSR timepoint. A similar comparison of the 50 most significant genes induced by the unfolded DD at the late timepoint (*Figure 4C*) reveals only three in common with the early HSR timepoint and 15 in common with the late timepoint (13 induced and 2 reduced).

To make the comparison in the reverse sense, comparison of the 50 most significantly induced genes at the early HSR timepoint (*Figure 4D*) reveals 48 in common with the late timepoint (all of

them induced). However, comparison of the top 50 early HSR responders with the early unfolded-DD timepoint finds only 8 genes in common (3 induced and 5 repressed), and comparison with the late unfolded-DD timepoint reveals 13 in common (2 induced and 11 repressed). Likewise, comparison of the 50 most significantly induced HSR genes at the late timepoint (135 min) with the genes induced by the unfolded DD finds 8 in common (7 induced and 1 repressed) with the earlier timepoint and 13 in common (10 induced and 3 repressed) with the later timepoint (*Figure 4E*). This lack of concordance between the genes induced by the unfolded DD and the genes induced by HS is suggestive, but certainly not conclusive, that the cellular-signaling pathways used to respond to these two perturbations are not identical.

If the HSR shared common signaling mechanisms with the cellular response to unfolded DD, then one would expect to see similar changes in gene-expression profiles (*Brannon et al., 2010*). When analyzing the responsive genes in an unbiased fashion, relying solely on the statistical significance of the observed changes, we see no such concordance (*Figure 4B–E*). Another way to approach the issue is to look at groups of genes that are known to share a common function. For example, cells experiencing proteotoxic stress might respond by producing more proteasomes to rid the cell of unfolded protein (*Vilchez et al., 2013*). When we looked at 39 genes that encode proteasome subunits, we found that the appearance of the unfolded DD causes changes in 16 of these genes, all of them up-regulated (*Figure 4F*). The HSR induces changes in only 3 genes, all of them down-regulated. The limited degree of overlap between these two sets of induced genes leads us to conclude that these two cellular stress responses are mechanistically distinct, though some degree of overlap cannot be excluded.

## Unfolded DD induces p53 accumulation

We quantified cellular levels of p53 protein following withdrawal of S1 by immunoblotting. The levels of p53 rose at 45 and 135 min, but p53 levels were elevated but falling by 405 min (*Figure 5A*). This result is consistent with the transcript analysis from both mRNA-seq and RT-qPCR in which the mRNA levels of p53 target genes such as *Mdm2* and *Cdkn1a* were strongly induced at 135 min but were lower by 405 min following the appearance of the unfolded DD (*Figure 2*).

To gain further insight into the role of p53 in response to unfolded DD, we used immunofluorescence analysis to quantify p53 levels and subcellular localization in the DD-GFP cells. Before withdrawal of S1, p53 levels were low and DD-GFP levels were high (*Figure 5B*). Levels of p53 rose by 45 min following the appearance of the unfolded DD and remained elevated through 405 min. Consistent with FACS analysis (*Figure 5—figure supplement 1A*), levels of DD-GFP remained relatively high through 135 min but significantly lower by 405 min. This experiment also shows that p53 is enriched in nuclei upon appearance of the unfolded DD, and statistical analysis of the immunofluorescence images supports these conclusions that p53 levels significantly rise and that p53 localizes to the nucleus (*Figure 5C*).

It is possible that p53 levels rise as a result of an indirect secondary effect caused by the unfolded DD. DNA damage is well known to activate p53 (*Kenzelmann Broz et al., 2013*), so we triggered the appearance of the unfolded DD and then immunoblotted cell lysates for various markers of DNA damage including phospho-KAP1, phospho-RNA32, phospho-Chk1, and γ-H2AX (*Figure 5—figure supplement 1B*). None of these hallmarks of DNA damage were significantly induced, suggesting that the DD-GFP-expressing cells were not experiencing DNA damage. ATM and ATR are kinases that are essential for cells to respond to DNA damage (*Cimprich and Cortez, 2008*), so we used chemical inhibitors of ATM and ATR to assess the potential role of these kinases in the cellular response to the appearance of unfolded DD (*Figure 5—figure supplement 1C*) (*Hickson et al., 2004*; *Couch et al., 2013*). Neither inhibitor affected the cellular response to the unfolded DD, suggesting that DNA damage signaling mediated by ATM or ATR is not responsible for the increased levels of p53. Oxidative stress can also induce p53, so we used CM-H2DCFDA as a sensor for reactive oxygen species in DD-mCherry-expressing cells upon withdrawal of S1. We observed no significant signal by analytical flow cytometry upon creation of the unfolded DD (*Figure 5—figure supplement 1D*). We conclude that cellular changes in p53 levels are a response to the unfolded DD and not to indirect sequelae such as DNA damage or reactive oxygen species.

To further validate the role of p53 in the cellular response to unfolded DD, we used RNA interference to knockdown p53 in DD-GFP-expressing cells. We then performed RT-qPCR to quantify

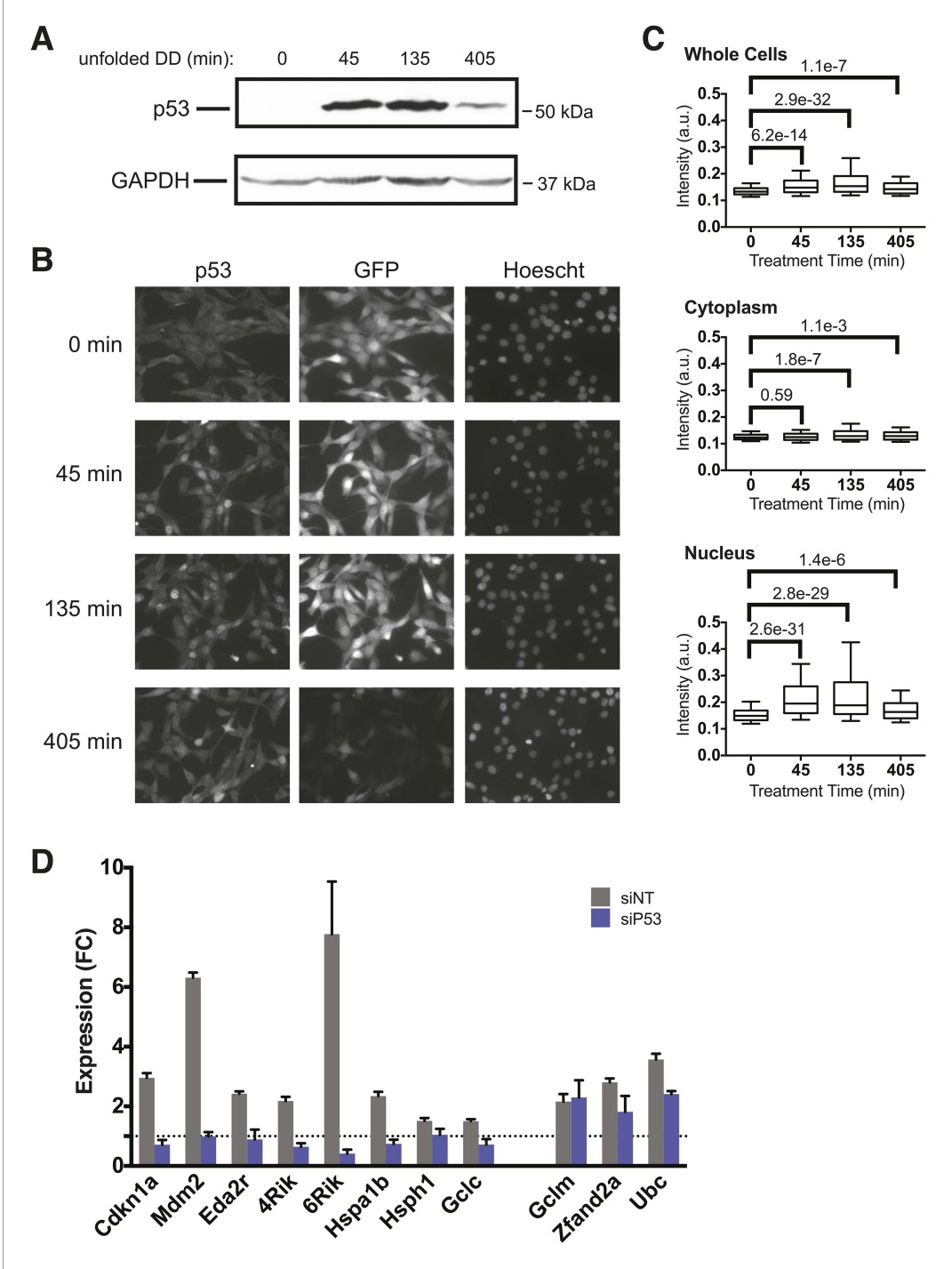

**Figure 5**. p53 protein accumulates upon appearance of the unfolded DD. (**A**) S1 was withdrawn from NIH3T3 cells expressing the DD-GFP fusion protein for the indicated times, and equal amounts of total protein were immunoblotted with antibodies against p53 or GAPDH. (**B**) S1 was withdrawn from cells expressing DD-GFP, and the cells were fixed at the indicated times. The levels of p53 and GFP were quantified using indirect immunofluorescence, and Hoechst 33,342 was used to image nuclei. (**C**) Quantification of the p53 signal observed in whole cells, cytoplasm, or nuclei from immunofluorescence data. The Welch test was used to calculate p-values. (**D**) Cells expressing DD-GFP stabilized by S1 were treated with siRNA pools against p53, cultured for 2 days, then RNA was harvested for RT-qPCR analysis 135 min following withdrawal of S1. Transcript levels for the indicated genes were normalized to GAPDH and expressed as fold-change relative to unperturbed cells. siNT = non-targeting control.

*Figure 5. continued on next page*

*Figure 5. Continued*

The following figure supplements are available for figure 5:

**Figure supplement 1**. Investigating potential sources of p53 activation.

**Figure supplement 2**. Involvement of other transcription factors.

---

changes in mRNA levels for various target genes following withdrawal of S1. Knockdown of p53 diminished the DD-dependent induction of p53 targets such as *Mdm2*, *Cdkn1a*, and *Eda2r* (*Figure 5D*). However, increased mRNA levels were not affected for some of the responsive genes including *Ubc*, *Zfand2a*, and *Gclm* (*Figure 5D*). These results suggest the involvement of other factors during the response to unfolded DD. Since we see induction of HS proteins and oxidative stress genes, we performed used RNAi to knockdown *Hsf1* and *Nrf2*, two transcription factors that mediate cellular responses to HS and oxidative stress. Cells were treated with siRNA reagents to knockdown these two genes and RT-qPCR was used to quantify mRNA levels of two dozen genes that are induced by the unfolded DD. Although most of the responsive genes were unaffected by knockdown of HSF1 and NRF2, loss of these factors did affect the induction of some responsive genes. Loss of Hsf1 attenuated the induction of several HS proteins (*Hspa1b*, *Hsph1*, *Dnajb1*) and loss of Nrf2 affected the induction of oxidative stress genes (*Hmox*, *Osgin*, *Sqstm1/p62*), respectively. These studies are far from comprehensive, but we can conclude that the cellular response to unfolded DD involves several stress-responsive transcription factors (*Figure 5—figure supplement 2*).

## Cellular responses differ for nuclear or cytosolic unfolded proteins

Cells have evolved protein QC surveillance mechanisms to monitor discrete cellular compartments such as the secretory pathway (erUPR) and mitochondria (mitoUPR), so we considered the possibility that the unfolded DD may elicit different transcriptional responses if unfolded protein were to appear in the nucleus or in the cytosol. We tagged the DD-GFP fusion protein with a nuclear localization signal and created an NIH3T3 cell line that expressed the DD-GFP fusion protein primarily in the nucleus. A complementary cell line was created, expressing a DD-GFP-mCherry fusion protein that was tagged with a nuclear export sequence. FACS-based cell sorting was used to select cells that express both the nucleus-localized and the nucleus-excluded DD-GFP at similar levels (*Figure 6—figure supplement 1A*). Fluorescence micrographs of both cell lines treated with S1 revealed that the targeted DD-GFP fusion proteins are present primarily in their expected compartments (*Figure 6A*). Analytical flow cytometry was used to quantify the rates of DD-GFP degradation upon withdrawal of S1, revealing that the nuclear DD-GFP was degraded with faster kinetics than the cytosolic DD-GFP (*Figure 6—figure supplement 1B*).

Unfolded DD was created in each cell line, and cells were harvested at 45, 135, and 405 min for transcript analysis using RT-qPCR. The mRNA levels of several genes, predominantly known p53 targets such as *Cdkn1a*, *Gadd45a*, and *Mdm2* respond to the appearance of either the nuclear or cytosolic unfolded DD (*Figure 6B*, *Figure 6—figure supplement 1C*). Other genes including DNA polymerase kappa and *Trp53inp1* are more sensitive to the unfolded DD in the nucleus. Conversely, we observed that transcript levels of many genes are relatively insensitive to the appearance of unfolded DD in the nucleus but are strongly induced by cytosolic unfolded DD (*Figure 6C*, *Figure 6—figure supplement 1C*). Genes falling into this group include chaperones (*Hsp90aa1*, *Hspa1b*, *Hspa8*, *Dnajb1*), proteasome subunits, known protein QC surveillance proteins (*Bag3*, *Zfand2a*, *Gclm*, *p62/Sqstm1*), and two of the four mammalian ubiquitin genes (*Ubb* and *Ubc*).

## The cellular response to unfolded DD provides a fitness benefit to dividing cells

In order to assess how cellular fitness might be affected by the appearance of the unfolded DD, we co-cultured the DD-GFP-expressing cells with unmodified NIH3T3 cells (*Sancho et al., 2013*). We created unfolded DD by withdrawing S1 for 45, 135, or 405 min. Cells were co-cultured for an additional 17 hr in S1-containing media and then analyzed by flow cytometry to quantify the ratio of DD-GFP-expressing cells

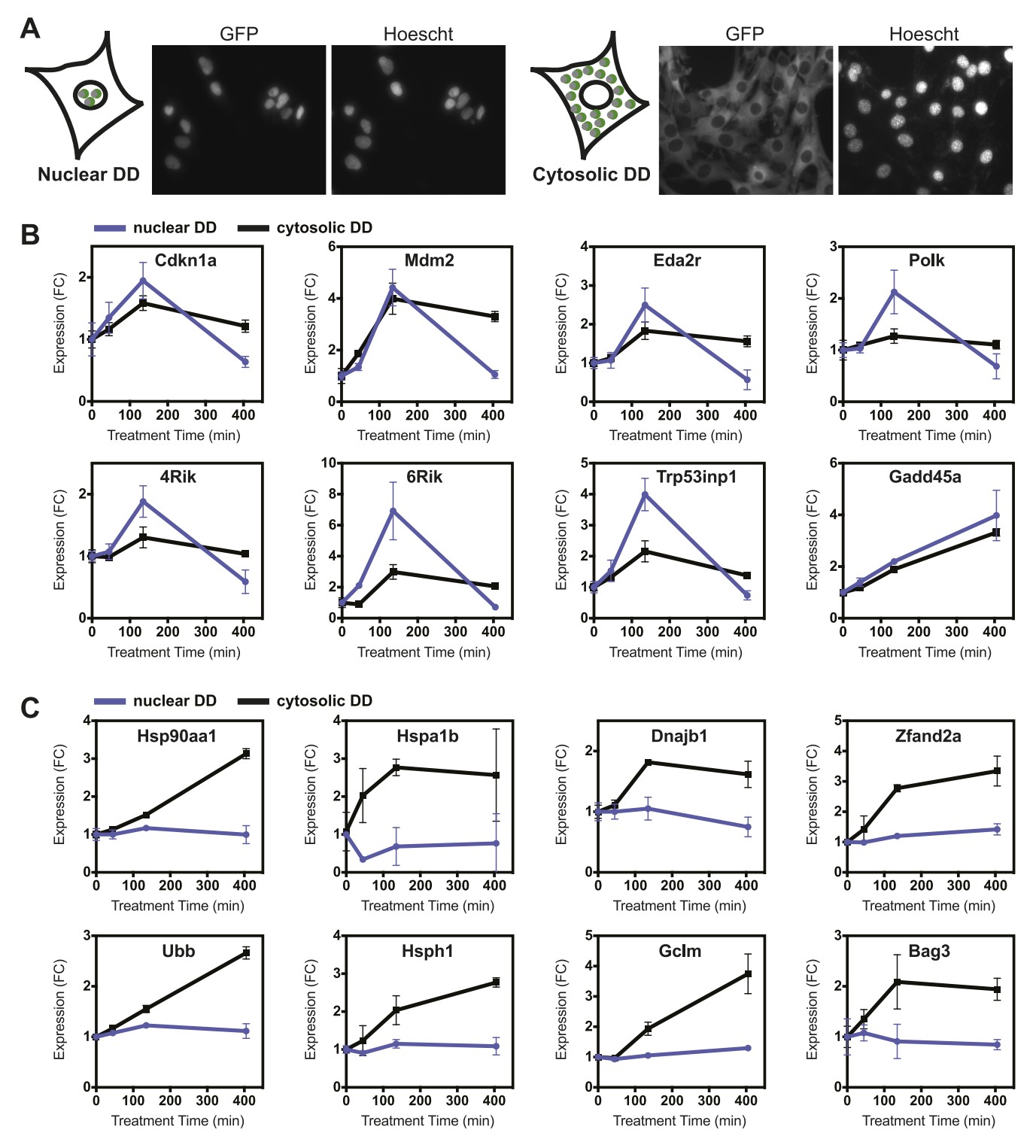

**Figure 6**. Nuclear or cytosolic unfolded proteins elicit distinct responses. (**A**) Images of live NIH3T3 cells stably expressing nuclear DD-GFP or cytosolic DD-GFP. Hoechst 33,342 was used to image nuclei. (**B**, **C**) Unfolded DD was induced by withdrawal of S1, and transcript levels of specific genes were quantified using RT-qPCR at the indicated times. Genes that respond more strongly to nuclear unfolded DD are shown in panel **B**, and genes that are induced by cytosolic unfolded DD are shown in panel **C**. Transcript levels were normalized to GAPDH and expressed relative to unperturbed samples.

The following figure supplement is available for figure 6:

**Figure supplement 1**. Additional comparisons of nuclear vs cytosolic unfolded protein stress.

to control cells (*Figure 7A*). We observed that the amount of time cells were exposed to the unfolded DD inversely correlated with replicative fitness. Creating unfolded DD for a period as brief as 45 min caused a significant and reproducible reduction in the population of DD-GFP cells, and 405 min of exposure to unfolded DD caused the original ratio of 61:39 (DD-GFP:control cells) to fall to 51:49. Cells that express threefold lower amounts of the DD do not experience this replicative fitness defect (*Figure 7A*). In attempt to identify the source of this replicative fitness defect, we used RNAi to knockdown p53, Hsf1, and Nrf2 in separate populations of cells. Loss of p53 reduces the replicative phenotype exhibited by cells challenged with the unfolded DD, whereas loss of Hsf1 or Nrf2 did not reveal any detectable effect in cycling cells (*Figure 7—figure supplement 1A–C*). Thus, there is a replicative cost for cells challenged with the unfolded DD, and this response appears to be mediated, at least in part, by p53.

We used a similar co-culture experiment to examine whether the cellular response to the appearance of unfolded DD protects cells from the adverse effects of unfolded protein. Cells expressing DD-GFP were mixed with NIH3T3 cells and co-cultured in the presence of S1. The S1 was then either withdrawn for 5 hr or S1 was maintained in the media. At the 5-hr timepoint, S1 was re-administered, and the cells were co-cultured for an additional hour to allow recovery from the unfolded DD. At this point, the co-cultured cells were then treated with either vehicle or with 30 μM sodium arsenite (*Yun et al., 2008*), cultured overnight, and analyzed at 24 hr (post appearance of unfolded DD) by flow cytometry. Both proteotoxic stresses affect the replicative potential of cells expressing the DD-GFP fusion protein. If the effects of the two stresses were additive, one could estimate an expected ratio of the DD-GFP:control cells (*Figure 7B*). Cells that responded to the appearance of the unfolded DD were strongly cross-protected from arsenite toxicity. A similar effect was seen when a different proteotoxic stress, the proteasome inhibitor MG132, was used to challenge the co-cultured cells (*Figure 7B*, *Figure 7—figure supplement 1D*). These results suggest that the transcriptional response induced by the appearance of

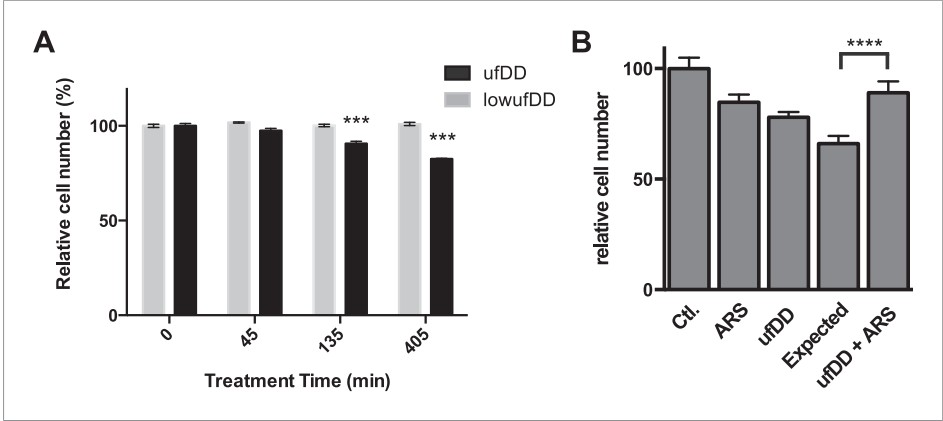

**Figure 7**. The cUPR provides protection against subsequent proteotoxic stress. (**A**) NIH3T3 cells expressing DD-GFP were co-cultured with unmodified 3T3 cells in the presence of 1 μM S1. The stabilizing S1 was withdrawn for 45, 135, or 405 min and then re-administered to the cells for an additional 17 hr. Analytical flow cytometry was used to quantify the relative populations of each cell type. The experiment was performed in triplicate for populations of cells expressing high levels of DD-GFP that induce the unfolded DD stress response (ufDD, black) as well as the cell line expressing threefold lower levels of DD-GFP (low ufDD, gray) as shown in *Figure 2B*. (**B**) Cells expressing DD-GFP were co-cultured with unmodified 3T3 cells in a 24-well plate. One group of cells was not exposed to stress from either the unfolded DD or 30 μM aqueous sodium arsenite (Ctl). The second group was exposed to arsenite only (ARS). The third group experienced the unfolded DD by withdrawal of S1 (shown as ufDD), and the fourth group was exposed to the unfolded DD followed by arsenite stress (ufDD + ARS). The relative populations of each cell type were quantified using analytical flow cytometry, with the control sample as the reference point for cells expressing DD-GFP. Expected = predicted fraction of DD-GFP cells assuming the unfolded DD and arsenite are additive. \*\*\* p-value < 0.001 by Welch's test, \*\*\*\* p-value < 0.0001 by Welch's test.

The following figure supplement is available for figure 7:

**Figure supplement 1**. The effect of transcription factors on cell growth.

the unfolded DD provides a fitness benefit by protecting cells against other forms of proteotoxic stress (*Calabrese et al., 2007*; *Morimoto, 2011*).

## Discussion

Our understanding of the cellular responses to protein-folding stress has been achieved through the use of relatively blunt experimental perturbations. Culturing eukaryotic cells at elevated temperatures or in the presence of hydrogen peroxide affects not only proteins but virtually every molecule in the cell (*Zunino et al., 2001*). Nevertheless, these perturbations have enabled many insights into the HSR and other protein-folding stress responses. Studies of the erUPR in the secretory pathway rely heavily upon the use of small molecules such as tunicamycin, thapsigargin, or dithiothreitol, and ethidium bromide is used to rapidly stress the protein-folding environment within mitochondria. Genetic methods to introduce stress are more specific but not as acute. Transgenes encoding folding-defective mutants of cellular proteins (e.g., CPY*) have been valuable for studies of protein folding and trafficking (*Knop et al., 1996*). However, there are few examples of mutant proteins being used to activate the HSR, despite the conventional wisdom that the HSR responds to unfolded proteins (*Wolff et al., 2014*). Mutant proteins that are prone to misfold affect cellular fitness when constitutively expressed (*Geiler-Samerotte et al., 2011*). However, the chronic nature of these genetically encoded stressors allows cells to adapt to their presence, rendering these perturbations unsuitable for studies of cellular responses that are induced by acute stress.

Given the success achieved using the acute, small molecule perturbants described above, we were optimistic that the ability to rapidly create a specific unfolded protein would illuminate cellular surveillance mechanisms that otherwise would be difficult to observe (*Chu et al., 2013*). The DDs we employed possess many attributes that make them attractive as conditional unfolded proteins. DDs are genetically encoded, providing opportunities to control subcellular localization. Additionally, DDs are maintained in a metabolically stable, folded conformation when bound to their stabilizing ligands, but withdrawal of the stabilizing ligand initiates the rapid unfolding of the DD (*Egeler et al., 2011*). Considered together, these features endow the DDs with the specificity inherent in genetic approaches combined with the temporal advantages derived from the use of small molecule pharmacology (*Rakhit et al., 2014*).

The cellular response to unfolded DD shares some functional similarities with the HSR and the erUPR. The transcriptional response is dose-dependent in the sense that lower levels of unfolded DD trigger a weak response or no response relative to higher levels of unfolded protein. Presumably, cells do not need to respond to the unfolded DD if they can degrade it rapidly, similar to the erUPR is not activated until endoplasmic reticulum-associated degradation is either overwhelmed orincapacitated (*Travers et al., 2000*). Additionally, as is observed with other stress responses, cells that mount the response to the unfolded DD pay a replicative fitness cost relative to cells that do not. However, a benefit accrues to cells that initiate this transcriptional response in the form of cross-protection from other forms of proteotoxic stress.

Despite these similarities, the cellular response to the unfolded DD and the HSR appears to be mechanistically distinct, although it is important to recall the challenges inherent in such a comparative analysis. First, the perturbations are fundamentally different. The discrete unfolded DD is qualitatively and quantitatively different from the variety of changes wrought by HS. HS will cause many proteins to unfold, and the overall amount of unfolded protein will likely be higher than can be attained using the unfolded DD. Additionally, HS causes hundreds or thousands of different proteins to unfold. This variety of stressors may present unique challenges for cellular QC surveillance pathways. Finally, HS affects all cellular molecules including nucleic acids and membranes.

Temporal differences present a second potential challenge for such a comparison. Given that the perturbations are fundamentally different in nature and intensity, the cellular response to two different stresses might begin at different times relative to the experimental initiation of the perturbations. Taken to its logical limit, even two identical cellular responses that are compared out-of-phase with each other (i.e., initiated at different times) may appear unrelated by gene expression analysis. So, lists of responsive genes should include any gene that responds to a stimulus in a statistically significant manner at any of the measured timepoints.

Even using such inclusive definitions for all three perturbations we employed in this study, the response to unfolded DD has little in common with the erUPR (*Figure 4A*). Genes that facilitate protein homeostasis in the secretory pathway may not necessarily be beneficial for protein folding in the cytosol or nucleus, so one might reasonably expect these two responses to be unrelated.

Conversely, given the shared biological goals of the cellular response to unfolded DD and the HSR, one might expect to find overlap in the lists of responsive genes. Gene expression analysis suggests that is the case: the number of induced genes found in common for both perturbations, though modest, is greater than would be expected by chance (*Figure 4A*).

Not surprisingly, several chaperone genes are induced by both the unfolded DD and the HSR including *Hsp90ab1*, *Hspa1b*, *Hsph1*, *Hspa8*, *Bag3*, *Dnajb1*, *Dnajb2*, *St13/Hip*, and *Hspb8*. However, many stress response genes are uniquely induced by the unfolded DD (*Supplementary file 2*). For example, the appearance of unfolded DD, but not the HSR, induces the expression of genes involved in aggresome formation such as *Hdac6*, *Vcp/p97*, several dynenin genes (*Dync1h1*, *Dync1li1*, *Dynll2*, *Dctn2*, *Dctn3*, and *Dctn4*), and the minus end-directed kinesins *Kif3c* and *Kifc1*. The unfolded DD uniquely induces genes involved in autophagy including *Ulk1*, *Tsc2*, *Wdr45*, *Atg2a* and *Atg2b*, *Vps11*, *Gabarapl1* and *Gabarapl2*, and cathepsins A and D. Many of the genes encoding proteasome subunits are induced by the unfolded DD but not the HSR (*Figure 4F*), and several other UPS genes are uniquely up-regulated by the unfolded DD including ubiquitin ligases (*Keap1*, *Ube4a*, *Ube4b*, *Huwe1*, *Klhl21*, *Ube2o*, *Ubr2*, *Ubr3*, *Ubr4*, *Herc2*, *Mdm2*) and deubiquitinating enzymes (*Usp14*, *Usp19*, *Usp20*, *Usp27x*, *Usp3*, *Usp5*). Finally, many apoptosis-related genes are uniquely induced by the appearance of unfolded DD including *Fas*, *Wrap53*, *Apaf1*, *Tnfrsf10b/DR5*, *Eda2r*, *Parp2*, *Parp4*, and *Bak1*.

One key difference between these two stress responses is the presence of a p53-dependent arm in the cellular response to unfolded DD. The involvement of p53 may explain many genes uniquely induced by unfolded DD, for example, the selective induction of the genes encoding proteasome subunits. However, mRNA-seq and ChIP-seq analyses of p53 target genes implicate only three of these genes (*Psmb3*, *Psmc3*, *Psmd13*) as direct p53 targets, out of the 16 that are uniquely induced by the appearance of unfolded DD (*Kenzelmann Broz et al., 2013*). Indeed, RNAi-mediated knockdown of p53 demonstrated that this transcription factor is involved in the induction of some, but not all, of the genes induced by unfolded DD (*Figure 5D*). Similar findings were observed when Hsf1 or Nrf2 were knocked down. This clear difference between the two cellular stress responses, not explained by these three stress-response pathways, suggests the involvement of at least one additional (not yet identified) arm of the cellular response to unfolded DD that is not shared by the HSR.

The biophysical properties of p53 may explain its absence from the HSR. At 37°C, the p53 core domain unfolds in vitro with a half-time of only 9 min, so it is possible that p53 is unable to maintain its folded, functional conformation when cells are exposed to the higher temperatures used to initiate the HSR (*Friedler et al., 2003*). If one were able to generate large amounts of unfolded protein at 37°C (i.e., simulate the HSR under standard culture conditions), p53 might be involved in the response. Indeed, treatment of mammalian cells with proteasome inhibitors, which causes misfolded proteins to accumulate, leads to increased p53 levels and the induction of p53-dependent proteins such as *Mdm2* and *Cdkn1*/p21 (*Lopes et al., 1997*; *Pandit and Gartel, 2011*).

Targeting the folded DD to either the nucleus or the cytosol allowed us to create unfolded protein in a specific cellular compartment. This unique experimental feature revealed that a subset of the genes induced by unfolded DD respond more strongly to nucleus-localized unfolded DD, whereas other responsive genes are more sensitive to unfolded cytosolic protein. The creation of unfolded protein in the nucleus induces primarily p53-dependent genes, implying that p53 (either directly or indirectly) may be responsible for protein QC surveillance in the nucleus. Gardner has characterized a nuclear E3 ubiquitin ligase in yeast called San1p that appears to recognize and ubiquitylate unfolded or misfolded proteins (*Rosenbaum et al., 2011*). Although SAN1 orthologs have not been found in higher eukaryotes, metazoans may likewise have evolved mechanisms to perform protein QC surveillance in the nucleus. The fact that p53 responds to the appearance of unfolded protein, especially its sensitivity to unfolded protein in the nucleus, suggests that p53 may play a central role in nuclear protein QC surveillance in addition to its role as the guardian of the genome. In support of this hypothesis, Sawa and co-workers have reported that mutant huntingtin protein binds directly to p53, leading to higher p53 levels and the induction of p53-responsive genes (*Bae et al., 2005*).

Although p53 appears to respond to unfolded protein that is created in either the nucleus of the cytosol, many of the genes that are induced by unfolded DD are strongly induced by cytosolic unfolded protein but respond modestly or not at all when the unfolded protein is created in the nucleus (*Figure 6C*). Several chaperones, proteasome subunits, ubiquitin genes, and redox-responsive genes fall into this category. This result suggests that there are one or more protein QC surveillance pathways that monitor unfolded protein in a compartment-specific manner.

The fact that cells respond so differently to the appearance of nuclear or cytosolic unfolded DD may have mechanistic implications for the cellular response to unfolded DD. We know that newly unfolded DD is rapidly ubiquitylated, which might deplete, at least temporarily, cellular levels of ubiquitin. One potential mechanism to initiate the cellular response to unfolded DD might include a futile cycle involving the production and rapid degradation of a transcription factor (*Barton and Sontag, 2013*). When conditions favor homeostasis, the surveillance transcription factor would be constitutively ubiquitylated and degraded. However, under conditions of proteotoxic stress, when ubiquitin becomes unavailable, the surveillance transcription factor would be stabilized and become competent to regulate the expression of genes involved in restoring homeostasis to the cell.

The mitoUPR operates in this fashion with ATFS-1 in the role of the surveillance factor (*Nargund et al., 2012*). Similarly, the Nrf2 transcription factor is degraded by the E3 ligase Keap1 until oxidative stress disables Keap1, allowing Nrf2 levels to rise and regulate redox-responsive stress genes (*Kobayashi et al., 2004*). A similar mechanism might explain the cellular response to unfolded DD, with p53 and other as yet unknown transcription factors in the roles of surveillance factors. However, one would expect the pool of cellular ubiquitin to rapidly partition between the nucleus and cytosol. Assuming rapid ubiquitin equilibration, the subcellular location of the unfolded protein should not affect the cellular response. The cellular response should be initiated by either cytosolic or nuclear unfolded protein, and the ensuing transcriptional programs should be identical. This is not what we observe, so it seems unlikely that ubiquitin depletion alone triggers the transcriptional response to unfolded DD.

The health of an organism depends, at least in part, on the ability of its protein QC surveillance mechanisms to ensure that misfolded proteins are degraded (*Chiti and Dobson, 2006*). There is evidence in humans that these surveillance systems become less effective with age, leading to higher levels of aberrantly folded proteins, particularly in post-mitotic cells such as neurons (*Douglas and Dillin, 2010*). Protein misfolding and aggregation is correlated with, and in some cases likely causative of, a variety of neurodegenerative diseases including Alzheimer's, Parkinson's, and Huntington's diseases (*Morimoto, 2008*; *Knowles et al., 2014*). It is well known that cells try to rescue aberrantly folded proteins by refolding them using chaperones or destroy them via the proteasome or autophagy (*Hartl et al., 2011*). However, we are only beginning to understand the basic mechanisms by which cells target cytosolic and nuclear proteins for degradation when the proper folded state cannot be achieved. One of the barriers to progress in this field is the lack of good tools and methods that would allow us to move beyond simple correlations to mechanistically probe the specific relationships between protein folding and the etiology of diseases where protein misfolding has been implicated. The use of the DDs to illuminate a new cellular response used to maintain protein homeostasis highlights how useful these molecular tools can be.

An increased understanding of the cellular response to the appearance of unfolded DD may also point to new strategies for diagnosing and ultimately treating unmet medical needs such as neurodegenerative diseases and cancer. Many features of this stress response remain to be discovered, including the molecular mechanisms that initiate the transcriptional response as well as the identities of the genes that contribute most strongly toward maintaining a healthy protein-folding environment (*Gibney et al., 2013*). Connections between protein QC surveillance mechanisms and disease are of particular interest, although hints are readily apparent. Mutations in proteins involved in the UPS have been implicated in neurological disease (e.g., Ube3a in Angelman syndrome and Ube3c in autism spectrum disorders) (*Greer et al., 2010*; *O'Roak et al., 2012*). The key sensor in the HSR, Hsf1, has recently been shown to regulate in malignant cancer cells a transcriptional program that is distinct from the program induced by HS (*Mendillo et al., 2012*). Finally, although a firm mechanistic picture is not yet clear, there is convincing evidence of an inverse correlation between Parkinson's disease and cancer, with a similar pattern becoming clear for Alzheimer's disease and cancer (*Driver et al., 2012*). The previously unappreciated role of p53 in protein QC surveillance might provide a mechanistic basis for this anticorrelation.

## Materials and methods

### Cloning

Constructs were subcloned using standard molecular biology techniques into the PiggyBac Transposon System vector pB (System Biosciences, Mountain View, CA), which the Super PB transposase recognizes transposon-specific inverted terminal repeat sequences located on both ends

of the transposon vector and moves the contents from the original sites and efficiently integrates them into TTAA chromosomal sites. We made plasmids with FKBP-derived DD-sfGFP, FKBP-derived DD-mCherry, FKBP-derived Nuclear DD-sfGFP, ecDHFR-derived DD-sfGFP, and FKBP-derived Cytosolic DD-sfGFP. For Nuclear DD-sfGFP, we placed the PPKKKRKV sequence, a nuclear localization signal from SV40 Large T-antigen, N-terminus of DD-GFP. Cytosolic DD-sfGFP was made by integrating nuclear exporting sequence from REV protein (LQLPPLERLYLD) in N-terminus and mCherry in C-terminus. Also, low-expressing DD-sfGFP and sfGFP constructs were cloned by the retroviral expression vector pBMN, which features long terminal repeats that contain all necessary elements for gene expression once integrated into the mammalian genome.

## Cell culture, transfections, and transductions

The NIH3T3 cell line (ATCC CRL-1658) was cultured in DMEM (Dulbecco's Modified Eagle Medium) supplemented with 10% heat-inactivated donor bovine serum (Invitrogen), 2 mM glutamine, 100 U/ml penicillin, and 100 μg/ml streptomycin. All other cell lines were cultured with 10% heat-inactivated fetal bovine serum (Invitrogen), 2 mM glutamine, 100 U/ml penicillin, and 100 μg/ml streptomycin.

Retrovirus was produced by transfecting ΦNX ecotropic cells with pBMN plasmids using standard TransIT-LT1 (Mirus) protocols. Viral supernatants were harvested 48 hr post-transfection and filtered (0.45 μm, Fisher). NIH3T3 cells were incubated with the retroviral supernatants supplemented with 4 μg/ml polybrene for overnight at 37°C. Cells were cultured in growth media for 48–72 hr to allow for viral integration. Infected cells were sorted by FACS for high-GFP level cells.

Cells were plated at $10 \times 10^4$ cells per well of a 6-well plate 12 hr prior to transfection. Cells were co-transfected with plasmid of pB vector and PiggyBac Transposase vector following standard protocols using TransIT-LT1 (Mirus). Cells were cultured in growth media for about a week for stable integration, and transfected cells were sorted by FACS for high-GFP level cells.

## Withdrawing ligand

To remove S1 from culture media for DD-GFP or GFP cells, we directly added purified 500 μM FKBP (F36V) protein in cultured media to make final concentration of 5 μM FKBP(F36V) protein. To withdraw trimethoprim from DHFRDD-GFP cells, cells were washed twice with culture media then incubated with the fresh media.

## Recombinant protein purification

BL21 cells expressing $His_6$-FKBP(F36V) protein (*Egeler et al., 2011*) or $His_6$-FKBP(F36V)-sfGFP protein were cultured in Luria–Bertani media at 37°C until optical density at 600 nm reached 0.5–0.7. Protein expression was induced with 0.5 mM isopropyl β-D-1- thiogalactopyranoside at 20°C overnight, then cells were harvested by centrifugation at 4500 RPM for 20 min at 4°C. Cells were lysed in phosphate-buffered saline (PBS) buffer supplemented with 1 mg/ml lysozyme (Invitrogen) and left on ice for 30 min. After sonication, Triton X-100 (Sigma) was added to 1%, and after 30 min incubation on ice, the solutions were centrifuged at 19,000 RPM for 30 min at 4°C. The soluble fraction was bound to Ni-NTA resin at 4°C for 1 hr. The resin was washed several times with wash buffer (50 mM $NaH_2PO_4$, 300 mM NaCl, 20 mM imidazole at pH 8) until eluent was free of protein by absorbance at 280 nm, then protein was eluted with elution buffer (50 mM $NaH_2PO_4$, 300 mM NaCl, 250 mM imidazole at pH 8). Eluted solutions were combined and dialyzed into dialysis buffer (PBS with 4% glycerol) and stored at −80°C.

## Flow cytometry

Cells were trypsinized and resuspended in culture media before flow cytometry experiments. Sorting was performed at the Stanford Shared FACS Facility using BD FACSAria sorter. For co-culture growth experiments, cells were trypsinized and resuspended in culture media before flow cytometry experiments. Then, we analyzed cells by FACSCalibur with no less than 25,000 events represented. Collected data were analyzed using FlowJo software.

## mRNA-seq

Total RNA was harvested using QIAShredder columns (Qiagen) and RNAEasy kit (Qiagen). Poly(A) RNAs were purified from total RNA using Dyna oligo(dT) beads (Invitrogen), and fragmented using RNA Fragmentation reagents (Life Technologies). For first-strand cDNA synthesis, we used fragmented RNA,

random hexamer primers (Life Technologies), 10xRT buffer (Life Technologies), 25 mM or 50 mM $MgCl_2$ (Life Technologies), 0.1 M DTT, RNaseOUT (Life Technologies), 10 mM dNTP mix (Life Technologies), and SuperScript III Reverse Transcriptase (Life Technologies) in 20 µl reaction volume. Samples were subjected to second-strand cDNA synthesis with DNA polymerase I (Invitrogen), 500 mM Tris-HCl pH 7.8 buffer, 50 mM $MgCl_2$ (Life Technologies), 0.1 M DTT, RNaseH (Life Technologies), 10 mM dNTP mix (Life Technologies). Samples were then subjected to end repair using End-It DNA End-Repair Kit (epicentre). We followed Illumina's genomic DNA library preparation protocol with some modifications. We used Ligafast (Promega) and a 1:40 dilution of the ligation adaptor mix for the adaptor ligation step. In addition, we used Encore SP+ Complete RNA seq kit (NuGEN) and Mondrian SP+ system (NuGEN) for preparing cDNA library for samples. The quality and amount of cDNA samples were checked by BioAnalyzer (Agilent) and Qubit (Life Technologies). Deep sequencing was performed with Illumina Hiseq 2000 in Stanford Sequencing Service Center. Sequencing data have been submitted to GEO (accession numbers GSE65504, GSE65636, and GSE71145).

## Bioinformatic analysis

Illumina reads were aligned to the mm9 mouse genome assembly and analyzed against RefSeq using the DNAnexus platform (https://dnanexus.com/). Any genes with read count of zero were removed from the sets. Then, differential gene expression analyses were performed by using edgeR, (http://www.bioconductor.org/). The lists of up- or down-regulated genes were obtained using a FDR of 0.05 as a cutoff. Gene enrichment analyses were performed using DAVID (http://david.abcc.ncifcrf.gov/). All statistical analyses were performed in R (www.rproject.org) or Prism 6 (http://www.graphpad.com/scientific-software/prism/).

## Hierarchical clustering and heatmap visualization

Hierarchical clustering with centered correlation and centroid linkage was implemented using Cluster 3.0 (http://bonsai.hgc.jp/~mdehoon/software/cluster/software.htm#ctv). Java TreeView software (jtreeview.sourceforge.net/) was used to create heatmaps.

## RT-qPCR

Total RNA was harvested using QIAShredder columns (Qiagen) and RNAEasy kit (Qiagen). We used 300 ng–1000 ng of total RNA, and treated samples with RQ1 RNase-Free DNase (Promega). For reverse transcription of RNAs, we used DNase-treated total RNA, a mixture of random hexamer primers (Life Technologies) and oligo dT primers (Life Technologies), 10xRT buffer (Life Technologies), 25 mM or 50 mM $MgCl_2$ (Life Technologies), 0.1 M DTT, RNaseOUT (Life Technologies), 10 mM dNTP mix (Life Technologies), and SuperScript III Reverse Transcriptase (Life Technologies) in 10 µl reaction volume. RT-qPCR analyses were carried out using Power SYBR Green PCR Master Mix (Life Technologies) on 7900HT Fast Real-Time PCR System (Applied Biosystems). Primer sequences are included in *Supplementary file 6*.

## Western blot analysis

Cells were washed with PBS and lysed with the buffer. Lysates were resolved by SDS-PAGE (Sodium Dodecyl Sulfate Polyacrylamide Gel Electrophoresis) and transferred to PVDF (Polyvinylidene Fluoride) membrane (Millipore) as previously described (*Egeler et al., 2011*). Detection was achieved using (Horseradish Peroxidase) HRP-conjugated secondary antibodies and Millipore Immobilon. Western blot quantification was performed using ImageJ (http://imagej.nih.gov/ij/).

## Antibodies

Antibodies were anti-GAPDH (mouse, 6C5, Abcam), anti-GFP (mouse, JL-8, Clontech), anti-HA (rat, 3F10, Roche), anti-tubulin (mouse, DM1A, Sigma), anti-p53 (rabbit, CM5, Vector Lab), anti-phospho-Ser824-Kap1 (rabbit, Bethyl Laboratory), anti-phospho-Ser345-Chk1 (rabbit, 133D3, Cell Signaling), anti-phospho-S4/S8-RPA32 (rabbit, Bethyl Laboratory ), anti- γH2AX (rabbit, 20E3, Cell Signaling), anti-phospho-Ser51-eIF2α (rabbit, Cell Signaling), and anti-Ub (mouse, VU-1, LifeSensors).

## Immunofluorescence

Cells were washed with PBS and fixed in 4% paraformaldehyde, permeabilized with 100 µg/ml digitonin (Sigma) in PBS and blocked with 2% BSA (Bovine Serum Albumin), 2% FBS (Fetal Bovine Serum) in PBS. Cells were incubated with 1:500 p53 CM5 antibody (Vector Lab) in PBS with 2% BSA, 2% FBS. After PBS washes, samples were incubated with 1:1000 Alexa 594-labeled secondary antibody (Invitrogen) and 2 µg/ml Hoechst 33,342 in PBS with 2% BSA, 2% FBS. After the second round of PBS washes, samples were mounted with ProLong Gold Antifade Reagent (Invitrogen). Images were captured on a Zeiss Axioskop 2 epifluorescence microscope equipped with a QICAM FAST 1394 digital CCD camera. Quantification of images was performed using CellProfiler (www.cellprofiler.org).

## siRNA experiments

We used these siRNA reagents: SMARTpool siGENOME Mouse Trp53 (M-040642-02-0005), SMARTpool ON-TARGET plus Mouse Hsf1 (L-040660-01-0005), SMARTpool ON-TARGET plus Mouse Nrf2 (L-040766-00-0005). siRNAs were transfected using Dharmafect1 and as described previously (*Chu et al., 2013*).

## Growth competition assay

Cells stably expressing DD-GFP and unmodified NIH3T3 cells were mixed and plated in 24-well plates. S1 ligand was withdrawn for various times. Cells were cultured for an additional 17 hr in S1-containing media and then analyzed by flow cytometry to quantify the ratio of DD-GFP-expressing cells to control cells. To assess protective benefits, we used a similar co-culture experiment to examine the potential for the appearance of the unfolded DD to provide a protective benefit to cells exposed to a second source of proteotoxic stress. Cells expressing DD-GFP were mixed with NIH3T3 cells and co-cultured in the presence of S1. The S1 was either withdrawn for 5 hr to trigger the cellular response or S1 was maintained in the media. At the 5-hr timepoint, a media change was used to re-administer S1 and the cells were cultured for an additional hour. At this point, cells were treated with either vehicle or with a second stress (30 µM sodium arsenite or 10 µM MG132), cultured overnight, and analyzed at 24 hr by flow cytometry. Cells that did not experience either stress were defined as the control and the ratio of $GFP^+:GFP^-$ cells at the end of the experiment was set to 100. The effects of the various treatments (unfolded DD only, arsenite or MG132 only, or both unfolded DD and arsenite or MG132) were then quantified using by flow cytometry. Expected effect of both stresses was calculated by multiplying the two mean values of either single-stress condition, and the error was computed by error propagation.

## Acknowledgements

We wish to thank Dan Jarosz for reading the manuscript and providing helpful advice as well as members of the Jarosz, Wysocka, Frydman, Cimprich, Meyer, Attardi and Elias labs for reagents and advice. We thank the Stanford Shared FACS Facility. This work was supported by the NIH (GM073046 and P50 GM107615), the Ellison Medical Foundation (AG-SS-2573-10), and the Stanford Discovery Innovation Fund. YM was supported by the Nakajima Foundation.

## Additional information

### Funding

| Funder | Grant reference | Author |
| --- | --- | --- |
| National Institutes of Health (NIH) | GM073046, P50 GM107615 | Yusuke Miyazaki, Ling-chun Chen, Bernard W Chu, Thomas J Wandless |
| Ellison Medical Foundation (EMF) | AG-SS-2573-10 | Yusuke Miyazaki, Ling-chun Chen, Bernard W Chu, Thomas J Wandless |
| Heiwa Nakajima Foundation | Graduate Student Fellowship | Yusuke Miyazaki |

The funders had no role in study design, data collection and interpretation, or the decision to submit the work for publication.

## Author contributions
YM, Conception and design, Acquisition of data, Analysis and interpretation of data, Drafting or revising the article; L-C, Acquisition of data, Analysis and interpretation of data; BWC, Conception and design, Acquisition of data, Analysis and interpretation of data; TS, TJW, Conception and design, Analysis and interpretation of data, Drafting or revising the article

# Additional files

## Supplementary files
• Supplementary file 1. List of genes significantly up- or down-regulated by the three stress conditions (false discovery rate [FDR] <0.05).

• Supplementary file 2. mRNA-seq data from all three perturbations.

• Supplementary file 3. Gene ontology analysis for genes responding to unfolded destabilizing domain (DD) stress.

• Supplementary file 4. Gene ontology analysis for genes responding to unfolded destabilizing domain (DD) stress in different timepoints.

• Supplementary file 5. mRNA-seq data from bortezomib perturbations.

• Supplementary file 6. List of primers used in RT-qPCR experiments in this paper.

## Major datasets
The following datasets were generated:

| Author(s) | Year | Dataset title | Dataset ID and/or URL | Database, license, and accessibility information |
|---|---|---|---|---|
| Miyazaki Y, Wandless TJ | 2015 | Cellular response to unfolded proteins in cytosol and nucleus | http://www.ncbi.nlm.nih.gov/geo/query/acc.cgi?token=kxsnkiqyhbyfdcb&acc=GSE65504 | Publicly available at the NCBI Gene Expression Omnibus (Accession no: GSE65504). |
| Miyazaki Y, Wandless TJ | 2015 | Cellular response to heat shock and ER stress | http://www.ncbi.nlm.nih.gov/geo/query/acc.cgi?token=grgnwomovbihhux&acc=GSE65636 | Publicly available at the NCBI Gene Expression Omnibus (Accession no: GSE65636). |
| Miyazaki Y, Wandless TJ | 2015 | Cellular response to proteasome inhibition by Bortezomib | http://www.ncbi.nlm.nih.gov/geo/query/acc.cgi?acc=GSE71145 | Publicly available at the NCBI Gene Expression Omnibus (Accession no: GSE71145). |

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
