## [Decision Letter]

Thank you for sending your work entitled “Distinct transcriptional responses elicited by unfolded nuclear or cytoplasmic protein in mammalian cells” for consideration at *eLife*. Your article has been favorably evaluated by Randy Schekman (Senior Editor), a Reviewing Editor, and two reviewers. After discussions between the reviewers, the Reviewing Editor has drafted this letter to help you prepare a revised submission.

In this interesting manuscript by Miyazaki and colleagues, they take advantage of an elegant technique to manipulate protein misfolding and turnover discovered in the Wandless lab. They use destabilizing domains regulated by exogenous ligands to cause immediate protein misfolding and turnover via proteasomes when the ligand (Shield 1) is removed from the media. These studies, in principle, add considerably to our understanding of how cells respond to misfolded proteins. Perhaps the strongest aspect presented here is the added temporal regulation that has not previously been possible in studies to study transcriptional adaptations to defective proteostatsis.

The authors utilize a destabilized domain (DD) of FKBP fused to GFP to evaluate the cellular response to the rapid accumulation of misfolded proteins in the cytosol or nucleus. Predominantly using transcriptional profiling approaches, the authors claim they identify a new stress-responsive signaling pathway, referred to as the cytosolic UPR (cUPR), which regulates cytosolic protein homeostasis in response to the rapid accumulation of a single misfolded protein. Furthermore, they show that the nuclear and cytosolic responses to the rapid accumulation of the unstable DD are different. While the general approach to monitoring cellular responses to misfolded proteins is creative and valuable, there are issues with the interpretation of the results and approach that need to be addressed prior to publication:

The removal of the pharmacologic chaperone from cells expressing the DD will result in the rapid unfolding of the DD and increased DD targeting to the proteasome. Thus, a consequence of the DD is increased proteasome load, potentially limiting proteasome activity. This mechanism appears to be reflected in the transcriptional profiling. To address this concern, the authors could perform a temporal mRNA-seq analysis of gene expression induced by the addition of proteasome inhibitors, as done for the DD stress, HSR and the erUPR. If the DD stress does mimic a proteasome inhibition (albeit likely a transient inhibition that is resolved following degradation of the accumulated DD), it would provide a mechanism of action to begin to define the transcriptional profile induced under this DD stress.

The above point highlights the lack of mechanistic follow-up of how the DDs challenge the cytosolic environment. Do the unfolded DDs inhibit the proteasome? Do DDs increase chaperone load? Is there activation of known cytosolic stress-responsive transcription factors such as HSF1 (likely for the induction of the cytosolic chaperones identified)? It is very difficult to identify new stress-responsive signaling pathways starting with a transcriptional profile, as proposed in this manuscript, in the absence of more mechanistic data on the type of proteome challenge and/or signaling pathway(s) involved. Without this type of information it is difficult to determine whether the observed transcriptional response reflects a new stress-signaling pathway, a physiologic level of activity of a known stress-signaling pathway, or a combination of multiple stress-signaling pathways activated in response to the unfolded DD. Thus, the claim that the observed transcriptional response in this manuscript reflects a new cUPR is not supported without identification of signaling pathways and/or mechanism of activation for DD induced stress. For instance, the HSR and UPR are largely defined by the transcripts induced by heat shock/tunicamycin in an HSF1/XBP1-dependent manner. While the authors are not expected to define all of the regulators of the described response, it would be helpful to know if it (or what fraction) is regulated by HSF1 (the data are convincing that it is not regulated by XBP1). It would be useful to perform these experiments in the absence of HSF1 (shRNA, CRISPR, deletion cells, etc) and determine if the transcript levels are altered. At least several genes should be examined by qRT-PCR to ensure that the transcripts identified here are not all/mostly regulated by HSF1.

Is the toxicity associated DD misfolding and turnover exacerbated/mitigated in cells lacking HSF1? This may further suggest the existence of separate pathways.

[Editors' note: further revisions were requested prior to acceptance, as described below.]

Thank you for resubmitting your work entitled “Distinct transcriptional responses elicited by unfolded nuclear or cytoplasmic protein in mammalian cells” for further consideration at *eLife*. Your revised article has been favorably evaluated by Randy Schekman (Senior Editor), a Reviewing Editor, and two reviewers. The manuscript has been improved but there are some remaining minor issues that need to be addressed before acceptance, as outlined below in Reviewer 1’s comments:

Reviewer #1:

The authors have addressed one of the major concerns in the original submission by demonstrating that inhibition of the proteasome leads to a transcriptional response that appears to be distinct from that observed following induced misfolding using the DD approach. Furthermore, the new experiments showing that HSF1 or NRF2 knockdown attenuate the induction of select DD-induced genes is also an important result for the revised manuscript. Collectively with the p53 knockdown results, it is clear that the DD activated an orchestrated combination of stress-responsive transcription factors to adapt cellular homeostasis to this specific type of insult. Overall, this is an interesting manuscript that describes how cells response to a specific misfolded protein in the cytosol and nucleus.

I have one note about Figure 7—figure supplement 1. In these experiments, you invert the starting amount of control 3T3 and DD-expressing cells (a 60:40 ratio). The other experiments show a 40:60 ratio. It would be nice to have some way to normalize all of these plots so they can be more easily compared. It is clear that this is a difficult experiment and it would be nice to easily visualize the amount of variability for these experiments. For example, it is clear in Figure 7 that the reduction in DD-expressing cells goes from ∼61–51%. Alternatively, in Figure 7—figure supplement 1 it goes from ∼61–55%. The effect with p53 knockdown is small, so having some way to determine the consistency of the experiment would be helpful. Statistical indicators would also be useful in Figure 7 and Figure 7—figure supplement 1 (in B and C to demonstrate the same reduction in fitness for cells with DD).

Also in the discussion of Figure 7 there is no mention of the low ufDD (black lines). I believe that shows that the effect is dosable (i.e. requires high level of DD). This is an interesting point that should be included.

---

## [Author Response]

We have performed several additional experiments to strengthen the manuscript, summarized as follows:

1) The original manuscript involved three perturbations (unfolded DD, heat shock, and treatment with tunicamycin) followed by mRNA-seq. As requested by the reviewers, we have treated the same cells with the proteasome inhibitor bortezomib and performed mRNA-seq at the same timepoints. There is little overlap in the set of genes induced by unfolded DD and bortezomib, and this experiment is included in the revised submission.

2) The original manuscript showed that some of the genes induced by the unfolded DD are regulated by p53 (Figure 5). We have done additional experiments using RNAi to knock-down HSF1 and Nrf2 followed by RT-qPCR to examine the potential role of these transcription factors in the cellular response to unfolded DD. These data are included in the revised manuscript.

3) Using a competitive co-culture experimental format, the original manuscript reported that the appearance of the unfolded DD caused a replicative fitness defect in the DD-expressing cells (Figure 7). We have now used RNAi to knock-down p53, HSF1, and Nrf2 in three different cell populations. Loss of HSF1 and Nrf2 have no effect on this fitness defect, however the lack of p53 appears to partially reduce the replicative stalling in response to the unfolded DD. These data are included in the revised manuscript.

[Editors' note: further revisions were requested prior to acceptance, as described below.]

We have made the following changes in accordance with the reviewers' suggestions:

1) Reviewer 1 suggests using “normalized” data for Figure 7 and the 3 panels of supporting information. My preference is always to show the raw data to provide maximal information to the reader, but I understand the desire to make the figure more “user-friendly” in this case. The best way to do this would be to calculate the selection coefficient for the cells experiencing the ufDD stress. However, since the appearance of the unfolded DD affects the rate of proliferation of the DD cells, one cannot honestly use the selection coefficient. We could use an “apparent selection coefficient”, but that's unacceptable in this case when we know the doubling rate is reduced for ufDD cells. So, in line with their suggestion, we have reformatted the figure to normalize the data to reflect how the percentage of cells that experience the ufDD change with respect to the control cells as a function of time spent challenged with unfolded DD. Figure 7 has been modified, as have been the 3 supporting panels showing similar data.

2) Those same 4 panels have been modified to include asterisk marks for significant conditions compared to time point 0 min.

3) In Figure 7, the cells expressing DD at 3-fold lower levels do not mount a strong response to the unfolded DD and do not suffer a fitness defect as a result. Our thanks to the reviewer for noticing that we did not address this fact in the text of the manuscript, and we have revised the manuscript accordingly.